# Pairwise Attribute Contrasting: Improving Zero-shot Image Classification of Vision-language Models

## Abstract

Vision-language models like CLIP have excelled in zero-shot inference by training on vast image-text datasets. However, relying solely on category names during inference limits their performance. Prior work introduced category descriptions generated by large language models (LLMs), aiming to enhance recognition and interpretability, albeit with challenges in capturing distinctions between fine-grained classes. We introduce Pairwise Attribute Contrasting (PAC), a zero-shot inference framework for vision-language models. PAC prompts LLMs to provide specific visual attributes that distinguish *category pairs*. To aggregate the pairwise comparisons into a single classification, PAC uses a voting procedure. Specifically, for each test image, all pairwise classifiers are first applied using their own pair-specific attributes to compute image-text similarities. A category receives a vote when it exhibits higher image-text similarity compared to the other class in the pair. Finally, the category that receives the highest vote becomes the final prediction. PAC shows consistent improvement on 18 benchmark datasets over other strong baselines across various model architectures. We further provide an efficient implementation by only computing text embeddings for unique attributes of a category, which significantly reduces the computation complexity compared to naively computing text embeddings for all attributes.

## 1 Introduction

Vision-language models, trained using contrastive approaches like CLIP (Radford et al., 2021), LiT (Zhai et al., 2022), and ALIGN (Jia et al., 2021) have recently achieved remarkable success. These models are typically trained on internet-scale datasets containing image-text pairs, such as LAION (Schuhmann et al., 2021). Thanks to the scale and diversity of training data, these models excel in conducting zero-shot inference across diverse downstream classification tasks while maintaining a high level of classification accuracy.

Unlike conventional deep neural networks, zero-shot classification involves a procedure where the similarity between test images and text prompts containing each category name is computed in the feature embedding space, and the category with the highest similarity is selected. In the standard zero-shot process, the sole reliance on category names to represent candidate categories often limits performance, as it heavily depends on the model's understanding of these names. Consequently, this approach tends to overlook other valuable information that could potentially enhance the recognition process. To mitigate this, prior work (Menon & Vondrick, 2022) leverages large language models (LLMs) to write category descriptions and average the image-text similarities over these descriptions in prediction. These descriptors generated by LLMs not only enhance the recognition performance but also offer a degree of interpretability regarding the model's prediction.

However, this method does not account for the inherent semantic similarities between categories and does not offer sufficient discriminative information between classes. When simply asking LLMs to write category descriptions, LLMs can generate general and similar descriptions for *different* categories that are *closely related in meaning*, as illustrated in Figure1(a). Thus, when using descriptions from LLMs as part of the text prompt, the model can still struggle to distinguish between such similar classes, which limits its performance on tasks like fine-grained classification.

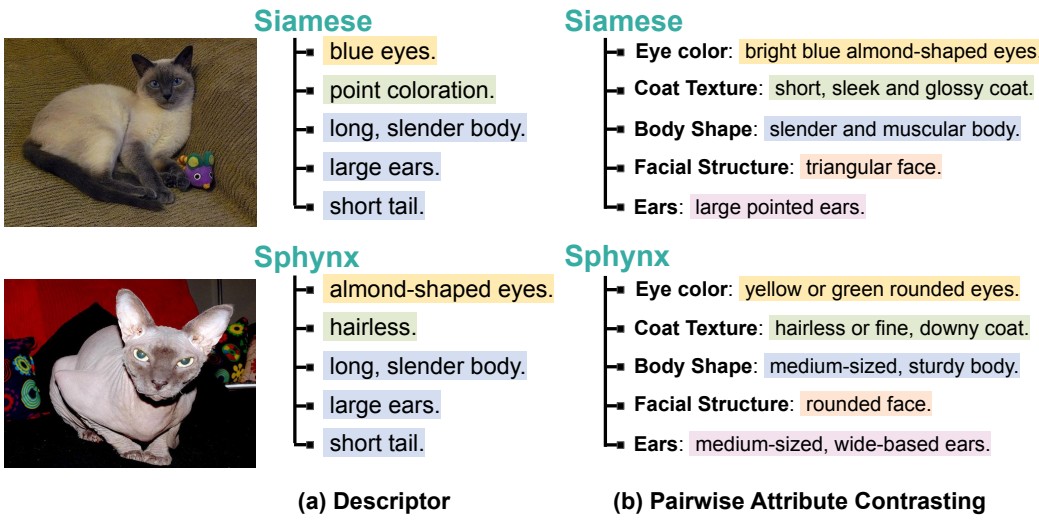

(a) Descriptor        (b) Pairwise Attribute Contrasting

Figure 1: **Comparison between Descriptor approach and PAC.** The descriptor approach (Menon & Vondrick, 2022) queries LLMs for object features but can lead to repetitive features (highlighted in the same color) for similar categories like Siamese and Sphynx cats. These redundant features hinder vision-language models from effectively distinguishing closely related categories. In contrast, PAC prompts LLMs to offer unique visual features, empowering models to leverage more discriminative information and enhance their performance in such cases.

In this paper, we present Pairwise Attribute Contrasting (PAC), a zero-shot inference framework for vision-language models. PAC addresses the aforementioned issue by prompting LLMs to provide specific visual attributes for distinguishing two categories at a time, rather than seeking general category descriptions from LLMs as in prior work (Menon & Vondrick, 2022). This way, LLMs can use their knowledge to explicitly generate pair-specific visual attributes to distinguish each pair of categories (Figure1(b)). When comparing different categories, LLM may vary the visual attributes it highlights. For instance, when comparing Sphynx and Siamese cats, LLM may emphasize attributes like *eye color* and *facial structure*, however, when distinguishing Sphynx cats from American Pit Bull Terriers, it utilizes *muzzle length* and *tail length* due to similarities in other features.

As the visual attributes provided by LLMs can vary when the same category is compared to different categories (as shown in Fig. 2), the conventional similarity-based zero-shot inference method (Radford et al., 2021) is not directly applicable to PAC. Hence, we redefine the prediction process as a voting procedure. When given a test image, PAC calculates image-text similarities for all pairs of categories using the corresponding pair-specific attribute descriptions from LLMs. Each class receives a vote when it exhibits higher image-text similarity compared to the other class in the pair. The class with the most votes becomes the prediction for the test image (see details in Fig. 3).

We conducted extensive evaluations of PAC on 18 benchmarking datasets for zero-shot evaluation of vision-language models. PAC consistently outperforms strong baseline methods by a large margin across various vision-language model architectures on the majority of these datasets. Moreover, as the pairwise mechanism in PAC results in quadratic complexity, we provide an efficient implementation which only computes text embeddings for unique attributes for each category. This approach significantly reduces computation complexity versus the naive approach of computing text embeddings for all attributes (details in Section 4.2).

## 2   RELATED WORK

**Vision-Language Models**. Vision-language models, which are trained on internet-scale image-text pairs, have achieved remarkable success in recent years. Models such as CLIP (Radford et al., 2021) have consistently demonstrated impressive zero-shot inference capabilities across various datasets.

Furthermore, the field has seen the emergence of other noteworthy models such as ALIGN (Jia et al., 2021), LiT (Zhai et al., 2022), FLAVA (Singh et al., 2022), FLORENCE (Yuan et al., 2021), and CoCa (Yu et al., 2022). In this work, our primary focus lies in enhancing the zero-shot performance of these vision-language models. To maintain consistency with prior studies and ensure the availability of extensive performance evaluations, we adopt CLIP as our primary model for assessment, given its well-established reputation for excellence in the field.

**Prompt Engineering for Vision-Language Models**. Previous research has explored methods to improve the performance of Vision-language models without the need for extensive fine-tuning on downstream datasets. These efforts have primarily followed two major directions: prompt tuning (Zhou et al., 2022b;a; Shu et al., 2022; Huang et al., 2022; Rao et al., 2022; Yang et al., 2023) and prompt crafting (Menon & Vondrick, 2022; Allingham et al., 2023; Novack et al., 2023). The key distinction between these two approaches lies in their methodologies. Prompt tuning involves the learning of contextual tokens, which are subsequently integrated into the prompts used for inference, which typically requires a training phase, either in supervised fashion (Zhou et al., 2022b;a; Rao et al., 2022; Yang et al., 2023; Rao et al., 2022; Yang et al., 2023) or unsupervised fashion (Shu et al., 2022; Huang et al., 2022). In contrast, prompt crafting involves the direct manipulation of text prompts, offering the advantage of not requiring any form of training. Moreover, it provides a certain level of interpretability in the process. ChiLS (Novack et al., 2023) focuses on improving coarse label scenarios by generating sub-classes from an existing hierarchy or by using LLMs. Our focus is instead on improving fine-grained scenarios.ZPE (Allingham et al., 2023) employs a scoring method involving weighted ensemble techniques with hand-crafted templates. In contrast, the Descriptor-based approach (Menon & Vondrick, 2022) relies on large language models to generate category descriptions, enriching their contextual understanding. Our method, PAC, falls under the category of prompt crafting and the key difference from prior work is the introduction of a novel pairwise attribute contrasting paradigm, which utilizes LLM-generated specific attribute descriptions for each category pair. This contrasts with previous methods like Menon & Vondrick (2022), which use more general descriptions.

## 3 Approach

### 3.1 Preliminaries

In the context of zero-shot image classification with vision-language models like CLIP (Radford et al., 2021), the primary objective is to assign one of the $N$ possible categories to a test image $x$, based on the knowledge of those category names. The essence of this task lies in establishing meaningful connections between the test images and the textual domain. The conventional methodology involves comparing image-text similarities in the feature embedding space and subsequently selecting the category with the highest similarity score.

A recent method (Menon & Vondrick, 2022) extends the usage of category names to category descriptions. Concretely, it begins by projecting the query image into a feature representation $I_x$. It then queries an LLM to generate $K$ descriptions for each category $i$. These descriptions are encoded into text embeddings $T_1^i, T_2^i, ... T_K^i$. To obtain the similarity score for a particular category, the model computes the average similarity between $I_x$ and each of these text embeddings: $s_i = \frac{1}{K} \sum_{j=1}^{K} T_j^i \cdot I_x$. Among all $N$ categories, the one with the maximum similarity score is predicted as the final class: $\hat{y} = \arg\max_i s_i$.

### 3.2 Pairwise Attribute Contrasting

Let us take a look at some of the descriptions produced by the LLM for two distinct categories: `Siamese cat` and `Sphynx` (Fig. 1 left). While there are two descriptions unique to each category (e.g., blue eyes vs almond-shaped eyes), there are also three which are same for both (e.g., large ears). Relying on those three descriptions will not give us anything useful to know whether an image belongs to `Siamese` or `Sphynx`.

We introduce our method - Pairwise Attribute Contrasting (PAC) - to solve this very problem. Our objective is to generate a set of descriptions of a category that are useful in separating it from every other category, in a pairwise fashion.

**Monkshood vs Daffodil**
- **Petal Shape**:
  - **Monkshood**: Spiked, helmet-shaped petals
  - **Daffodil**: Trumpet-shaped, flared petals
- **Flower Color**:
  - **Monkshood**: deep purple or blue
  - **Daffodil**: yellow or white
- **Leaf Shape**:
  - **Monkshood**: palmate, deeply lobed leaves
  - **Daffodil**: Linear, strap-like leaves

**Monkshood vs Geranium**
- **Flower Shape**:
  - **Monkshood**: Spiky, helmet-shaped blooms
  - **Geranium**: Rounded, cup-shaped flowers
- **Flower Color**:
  - **Monkshood**: deep purple or blue
  - **Geranium**: vibrant pink or red blossoms
- **Leaf Texture**:
  - **Monkshood**: smooth, glossy leaves
  - **Geranium**: crinkled, textured foliage

**BMW M3 Coupe vs Audi S4 Sedan**
- **Wheel Design**:
  - **BMW M3**: multi-spoke black alloy wheels
  - **Audi S4**: Five-spoke silvery alloy wheels
- **Grille Design**:
  - **BMW M3**: wide, low-profile double kidney grille
  - **Audi S4**: single-frame chrome grille
- **Headlight Shape**:
  - **BMW M3**: Round and compact headlights
  - **Audi S4**: Sleek and angular headlights

**BMW M3 Coupe vs BMW X5 SUV**
- **Body Style**:
  - **BMW M3**: sleek, two-door sports car
  - **BMW X5**: Robust, four-door SUV
- **Grille Design**:
  - **BMW M3**: wide, low-profile double kidney grille
  - **BMW X5**: tall, vertical double kidney grille
- **Roofline**:
  - **BMW M3**: Sloping, coupe-like roofline
  - **BMW X5**: single or dual exhaust outlets

Figure 2: **Example of Pair-specific Attribute Descriptions**.When comparing the same category to different categories, shared attributes (e.g., flower color and grille design, highlighted in the same color) exist, but other attributes vary (highlighted in various colors). This enables the vision-language model to employ distinct visual attributes tailored for distinguishing specific category pairs, rather than relying solely on general descriptions.

Specifically, instead of prompting LLMs with inquiries which are about a category in isolation, e.g., *What are useful features for distinguishing a category name in a photo?* (as done by the previously described method Menon & Vondrick (2022)), PAC explicitly requests LLMs to provide visual attributes capable of distinguishing *pairs of categories*:

```
Q: What visual attributes can differentiate between {category i}
   and {category j}?
A:
  - Attribute 1:
    - Category i: Description for Attribute 1
    - Category j: Description for Attribute 1
  - Attribute 2:
    - Category i: Description for Attribute 2
    - Category j: Description for Attribute 2
  ...
```

We probe the LLM in this way to generate the pair-specific attribute descriptions for all possible category pairs in the dataset. Importantly, it's worth noting that the visual attributes (and their corresponding descriptions) may vary when pitching the same category against two different ones. This allows vision-language models to leverage visual attributes specifically tailored for distinguishing a pair of categories, instead of general descriptions. For example, as shown in Figure 2, petal shape and leaf shape are used to differentiate *Monkshood* and *Daffodil* (two types of flowers) but are not used to differentiate *Monkshood* and *Geranium*. Similarly, wheel design and headlight shape are used to differentiate *BMW M3* and *Audi S4* but not used to differentiate *BMW M3* and *Audi X5*.

To conduct zero-shot classification with a vision-language model, we need to transform the pair-specific attribute descriptions into text prompts. We adopt the format `{category_name} with ( {attribute_name} {description})` to construct the prompt for each attribute description. This format is determined based on a single dataset and maintained consistently across all evaluation datasets. The only exception is for datasets with coarse labels, where we omit `{attribute_name}` from the text prompt. This adjustment is made because the attribute names generated by an LLM tends to be overly generic in such cases. For further insights into alternative template choices, please refer to Section 4.2.

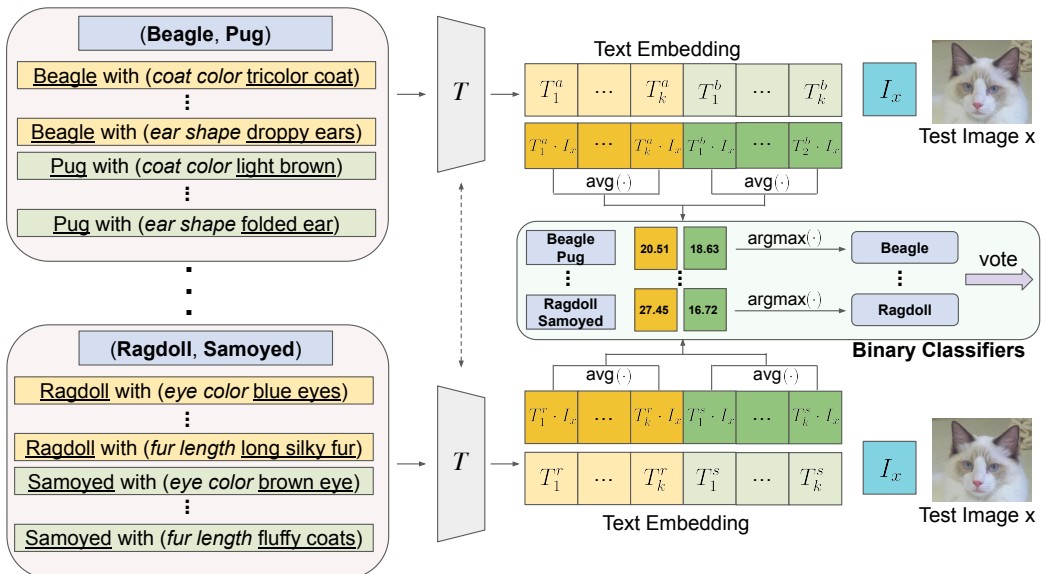

Figure 3: **Pipeline of Pairwise Attribute Contrasting (PAC) for zero-shot classification.** For each pair of categories (two categories are denoted in yellow and green), PAC leverages pair-specific attribute descriptions from an LLM and computes the text-image similarities between the image embedding and each text embedding. The binary classifier averages these similarities within each category and votes for the category with the higher averaged similarity. In this example, the first classifier favors Beagle because it exhibits greater similarity over Pug (20.51 vs. 18.63), while the second classifier votes for Ragdoll for the same reason. The votes from all binary classifiers are aggregated, and the category with the highest vote becomes the final prediction.

## 3.3 PAC FOR MULTI-CLASS CLASSIFICATION

The previous section described the process of creating the dataset of attribute descriptions for all pairs of categories in the dataset. Now, to classify an image as belonging to one of the categories, we take inspiration from the One-Versus-One (1v1) SVM framework in classic machine learning literature (Cortes & Vapnik, 1995; Burges, 1998; Sain, 1996). In 1v1 SVM, multiple binary classifiers are trained, one for each pair of categories, allowing them to effectively distinguish between those categories. Similarly, in our PAC formulation, we view the final classification problem as a series of one-versus-one binary classification problems.

Specifically, we create a binary classifier for every category pair $c_i$ and $c_j$ in the dataset, whose goal is to predict whether the test image $x$ is more likely to be category $c_i$ or $c_j$. The prediction for a category pair, $\hat{y}_{ij}$, is made as follows:

$$\hat{y}_{ij} = \begin{cases} c_i, & \text{if } s(c_i, I_x) > s(c_j, I_x) \\ c_j, & \text{otherwise} \end{cases} \tag{1}$$

where $s(c, I_x)$ is a similarity function defined as follows:

$$s(c, I_x) = \frac{1}{K} \sum_{a \in A(c)} \phi(a, I_x) \tag{2}$$

where $A(c)$ denotes the set of text embeddings corresponding to the $K$ attribute descriptions for category $c$, and $\phi(\cdot)$ computes the cosine similarity. The final category prediction is decided through a majority voting mechanism. The category corresponds to $\hat{y}_{ij}$ receives a vote and the category that receives the most vote will be selected as the prediction $\hat{y}$ for image $x$:

$$\hat{y} = c_k, \quad \text{where } k = \arg\max_m \left( \sum_{i=1}^{N} \sum_{j=1, j\neq i}^{N} \mathbb{1}\{\hat{y}_{ij} = c_m\} \right) \tag{3}$$

This whole pipeline is described in Fig. 3, where we see the process of classifying an image of *Ragdoll* (a breed of cat). The process consists of many binary classifications, of which we have shown two. In (i) *Beagle* vs *Pug* and (ii) *Ragdoll* vs *Samoyed*, *Beagle* and *Ragdoll* win out respectively and each gets a vote. However, across all the pair ups, *Ragdoll* wins the most and is ultimately given out as the (correct) prediction, as shown in the right figure.

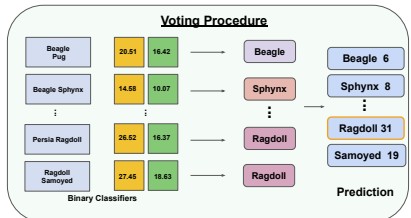

## 3.4 REDUCING COMPUTATION COMPLEXITY

As one may notice, the introduction of pair-specific attributes involves the computation of text embeddings and image-text similarities for each pair of categories in the dataset. Considering $N$ categories and $K$ attribute descriptions per pair, the total number of operations is given by $\binom{N}{2}K = \frac{N \cdot (N-1)}{2}K$, resulting in quadratic complexity with respect to $N$. In contrast, descriptor-based approaches demand $N \cdot K$ operations.

We optimize the computational efficiency of PAC by avoiding redundant computations of text embeddings and image-text similarities for attributes that are present across multiple pairs. As depicted in Figure 2, when a category is compared to the other $N-1$ categories, some attributes can be shared (e.g., flower color in the top example and grille design in the bottom example). We aggregate unique attributes for each category across different pairs and compute text embeddings and similarities only once for each attribute. During prediction with the binary classifier for pair $c_i$ and $c_j$, we utilize the text embeddings specific to this pair. We show in Section 4.2 that this practice significantly reduces the computation overhead compared to naively computing text embeddings and image-text similarities for every attribute.

## 4 EXPERIMENTS

In this section, we demonstrate the effectiveness of the Pairwise Attribute Contrasting (PAC) approach through a comprehensive set of experiments conducted across 18 datasets, spanning multiple vision-language model architectures. We categorize the datasets into two groups: fine-grained datasets and ImageNet variations. For all of our experiments, we use *gpt-3.5-turbo* as our LLM to generate pair-specific attributes. We also include results using other LLMs in Appendix A.1.

**Fine-grained Datasets**. Following zero-shot evaluation protocols from prior work (Radford et al., 2021; Allingham et al., 2023), we consider 11 fine-grained datasets for evaluation: Caltech-101 (Fei-Fei et al., 2004), CIFAR-10/100 (Krizhevsky et al., 2009), Stanford-Cars (Krause et al., 2013), Describable Textures Dataset (Cimpoi et al., 2014), EUROSAT (Helber et al., 2019), FOOD-101 (Bossard et al., 2014), Oxford-Flowers (Nilsback & Zisserman, 2008), Oxford-Pets (Parkhi et al., 2012), RESISC-45 (Cheng et al., 2017), and SUN397 (Xiao et al., 2016). We further consider CUB-200 (Wah et al., 2011) and PLACES-365 (Zhou et al., 2017) as used by Menon & Vondrick (2022). These together sum to 13 fine-grained datasets for evaluation.

**ImageNet-Family**. We further consider evaluation on ImageNet (Russakovsky et al., 2015) and its variations including: ImageNet-V2 (Recht et al., 2019), ImageNet-Sketch (Wang et al., 2019), ImageNet-Rendition (Hendrycks et al., 2021a), and ImageNet-Adversarial (Hendrycks et al., 2021b), which together form 5 datasets for evaluation.

**Baseline Methods**. We compare PAC with state-of-the-art methods that aim to improve the zero-shot accuracy of vision-language models. These include standard CLIP (Radford et al., 2021), descriptor-based techniques (Menon & Vondrick, 2022), and the ZPE approach (Allingham et al., 2023). The descriptor-based method utilizes LLMs to extract informative text descriptions for object recognition, while ZPE introduces a novel scoring mechanism that combines multiple handcrafted prompt templates through a weighted averaging procedure. Following previous research (Menon & Vondrick, 2022; Allingham et al., 2023), we conduct our evaluation with different model architectures from CLIP (Radford et al., 2021).

| | **Food-101** | | | **Describable Textures** | | | **Oxford-Pets** | | |
|---|---|---|---|---|---|---|---|---|---|
| | ViT-B/16 | ViT-L/14 | ViT-L/14@336 | ViT-B/16 | ViT-L/14 | ViT-L/14@336 | ViT-B/16 | ViT-L/14 | ViT-L/14@336 |
| CLIP (Radford et al., 2021) | 85.61 | 91.79 | 92.23 | 43.72 | 51.33 | 52.39 | 81.88 | 88.25 | 88.20 |
| Descriptor (Menon & Vondrick, 2022) | 88.50 | 92.44 | 93.26 | 45.59 | 54.36 | 54.95 | 86.92 | 92.23 | 91.69 |
| Descriptor* (gpt-3.5 re-implementation) | 88.31 | 92.95 | 93.71 | 47.12 | 55.85 | 56.96 | 87.47 | 91.37 | 91.52 |
| PAC (Ours) | **89.04** | **93.33** | **94.05** | **47.61** | **56.43** | **57.18** | **87.95** | **93.02** | 92.96 |

| | **CUB-200** | | | **EUROSAT** | | | **Places-365** | | |
|---|---|---|---|---|---|---|---|---|---|
| | ViT-B/16 | ViT-L/14 | ViT-L/14@336 | ViT-B/16 | ViT-L/14 | ViT-L/14@336 | ViT-B/16 | ViT-L/14 | ViT-L/14@336 |
| CLIP (Radford et al., 2021) | 56.35 | 63.08 | 63.41 | 43.36 | 41.48 | 44.80 | 38.27 | 39.00 | 39.58 |
| Descriptor (Menon & Vondrick, 2022) | 57.75 | 63.46 | 65.26 | 48.82 | 48.66 | 48.74 | 40.34 | 40.55 | 41.18 |
| Descriptor* (gpt-3.5 re-implementation) | 57.23 | **63.96** | 65.29 | 47.53 | 49.18 | 48.36 | **41.76** | 41.52 | 41.97 |
| PAC (Ours) | **58.13** | **63.96** | **65.34** | **49.53** | **50.57** | **51.71** | 41.14 | **41.60** | **42.14** |

| | **Oxford-Flowers** | | | **Stanford-Cars** | | | **RESISC-45** | | |
|---|---|---|---|---|---|---|---|---|---|
| | ViT-B/16 | ViT-L/14 | ViT-L/14@336 | ViT-B/16 | ViT-L/14 | ViT-L/14@336 | ViT-B/16 | ViT-L/14 | ViT-L/14@336 |
| CLIP (Radford et al., 2021) | 64.41 | 73.54 | 73.68 | 62.24 | 74.44 | 75.35 | 58.31 | 65.08 | 65.92 |
| Descriptor* (gpt-3.5 re-implementation) | 70.35 | 75.85 | 75.28 | 63.41 | 74.71 | 76.25 | 58.85 | 64.98 | 65.62 |
| PAC (Ours) | **70.97** | **76.25** | **76.69** | **63.53** | **75.46** | **77.12** | **59.35** | **66.16** | **66.57** |

Table 1: **Zero-shot Classification Accuracy without Hand-crafted Templates**. CLIP baselines are evaluated using class names only and Descriptor uses the format {*category*} *which (is/has/etc)* {*descriptor*}. Likewise, our approach PAC uses the format {*category*} *with* ({*attribute*} {*description*}) in its pairwise attribute contrasting procedure.

| | CALTECH | CARS | C10 | C100 | DTD | EURO | FOOD | FLOWERS | PETS | RESISC | SUN |
|---|---|---|---|---|---|---|---|---|---|---|---|
| CLIP (Radford et al., 2021) | 82.82 | 64.17 | 89.10 | 65.90 | 45.64 | 51.60 | 88.66 | 71.23 | 88.91 | **65.44** | 63.87 |
| ZPE (Allingham et al., 2023) | 85.54 | 64.62 | 89.30 | 66.63 | 46.28 | 53.82 | 88.61 | 70.17 | 88.72 | 64.22 | 64.70 |
| PAC (Ours) | **86.29** | **64.83** | **89.81** | **67.51** | **49.46** | **57.91** | **89.22** | **72.56** | **89.88** | 65.06 | **67.17** |

Table 2: **Zero-shot Classification Accuracy on Fine-grained Datasets with CLIP ViT-B/16**. Results are obtained with hand-crafted prompt templates. CLIP baselines are evaluated using the average of hand-crafted prompt templates. ZPE designs a weighting mechanism across a large pool of 247 hand-craft templates, while our approach PAC is evaluated over the top-10 templates found by ZPE to reduce computation overhead.

## 4.1 RESULTS

**PAC outperforms Descriptor-based Techniques**. To ensure a fair comparison with prior work (Menon & Vondrick, 2022), we conduct experiments without relying on any dataset-specific hand-crafted prompt templates. In this context, Menon & Vondrick (2022) employs text prompts in the format `{category_name} which (is/has/etc) {descriptor}`, establishing connections between category names and descriptions. Similarly, PAC uses the format `{category_name} with ({attribute_name} {description})`. Importantly, these standardized text prompts are applicable across all datasets, marking a distinction from the dataset-contextualized hand-crafted prompt templates in other studies (Radford et al., 2021; Zhai et al., 2022).

Table 1 demonstrates the significant advantages of our approach, PAC, over the previous method (Menon & Vondrick, 2022) across multiple CLIP visual encoder architectures. Even with the best-performing model (ViT-L/14@336), PAC consistently achieves significant improvements. For instance, on Describable Textures and EUROSAT datasets, PAC demonstrates an absolute accuracy improvement of 2.23% and 2.97% over reported results in Menon & Vondrick (2022). We further reimplement the descriptor approach using *gpt-3.5-turbo* for fair comparison and PAC can still outperform it by a large margin on most datasets. These findings underscore the effectiveness of our pairwise attribute contrasting approach.

**PAC also outperforms existing baselines on fine-grained datasets**. We further conduct evaluations on 11 fine-grained datasets. We extend our comparision to include ZPE (Allingham et al., 2023), a score-based ensemble mechanism based on an extensive set of 247 hand-crafted prompt templates. To demonstrate that PAC is also compatible with hand-crafted templates with minimal computational overhead, we simply select the top-10 hand-crafted templates with the highest weights as determined by ZPE and ensemble over these templates for each attribute description by averaging the corresponding text embeddings.

As shown in Table 2, with CLIP ViT-B/16 architecture, PAC consistently outperforms the CLIP and ZPE baselines except on RESISC. Notably, for the Flowers, DTD, EUROSAT, and SUN397 datasets, PAC exhibits a substantial improvement of 2-3% in absolute accuracy, surpassing other approaches

| | | IMAGENET FAMILY | FINE-GRAINED | ALL |
|---|---|---|---|---|
| CLIP ResNet-50 | CLIP (Radford et al., 2021) | 45.91 | 59.36 | 55.15 |
| | ZPE (Allingham et al., 2023) | 46.28 | 59.64 | 55.46 |
| | PAC (Ours) | **47.94** | **61.31** | **57.13** |
| CLIP ResNet-101 | CLIP (Radford et al., 2021) | 51.35 | 62.33 | 58.90 |
| | ZPE (Allingham et al., 2023) | 51.65 | 62.66 | 59.21 |
| | PAC (Ours) | **52.76** | **64.46** | **60.80** |
| CLIP ViT-B/16 | CLIP (Radford et al., 2021) | 60.85 | 70.67 | 67.29 |
| | ZPE (Allingham et al., 2023) | 61.21 | 71.15 | 67.73 |
| | PAC (Ours) | **62.51** | **72.70** | **69.51** |
| CLIP ViT-L/14 | CLIP (Radford et al., 2021) | 72.45 | 77.40 | 75.85 |
| | ZPE (Allingham et al., 2023) | 72.74 | 77.67 | 76.13 |
| | PAC (Ours) | **73.46** | **78.59** | **76.99** |

Table 3: **Zero-shot Classification Accuracy across Different Visual Encoders**. We evaluate PAC over four different CLIP visual encoders on 5 ImageNet variations and 11 fine-grained datasets. PAC consistently outperforms existing approaches across datasets with different visual encoders.

by a significant margin. These results underscore the effectiveness of PAC, when applied alongside hand-crafted prompt templates.

**PAC consistently improves over existing baselines across multiple architectures**. Finally, we present evaluations of PAC with other CLIP architectures on ImageNet family formed by 5 ImageNet variations (ImageNet-val, ImageNet-V2, ImageNet-Sketch, ImageNet-Rendition and ImageNet-Adversarial) and fine-grained datasets as shown in Table 3 (11 of them). Due to space limitations, we report the average accuracy of each group and the overall average accuracy across these 16 datasets. The accuracy on individual datasets can be found in Appendix A.2.

In Table 3, we also present the evaluation results obtained using various visual encoders, namely ResNet-50, ResNet-101, ViT-B/16, and ViT-L/14 within the CLIP framework. PAC consistently outperforms existing approaches by a substantial margin across all dataset categories and achieves impressive average performance improvements. Specifically, when employing the aforementioned encoder architectures, PAC demonstrates average accuracy improvements of 1.98%, 1.59%, 1.78%, and 0.86%, respectively. These results demonstrate that the performance gains achieved by PAC can generalize to different visual encoders and datasets spanning various contexts.

## 4.2 ABLATION STUDY

**Formulation of Text Prompts.** We present the results of employing different formulations to convert pair-specific attribute descriptions into text prompts, as outlined in Table 4. It's worth noting that the ideal formulation may vary from one dataset to another. While customizing the formulation for each dataset could potentially improve PAC's performance, we maintain a consistent choice. We use the same formulation initially adopted for Oxford-Pets when evaluating on multiple datasets. Given the absence of a validation set in zero-shot classification scenarios, this approach to text prompt formulation selection ensures the integrity of our evaluation process and upholds fairness.

The only exception is for datasets with coarse labels such as EUROSAT (Helber et al., 2019) and RESISC (Cheng et al., 2017) because the attribute names found by LLMs are typically too generic and do not make a lot of sense. Therefore, we simply remove `{attribute_name}` from the formulation and all other parts remain the same. We include few examples of pair-specific attribute descriptions of such datasets in Appendix A.3.

**Computation Complexity**. As discussed in Section 3.4, PAC's inference process involves pairwise binary classification, resulting in quadratic complexity with respect to the number of categories. Our efficient implementation only computes text embeddings and image-text similarities for unique attributes of each category, which leads to a substantial reduction in computational complexity. As illustrated in the right figure, this approach significantly reduces the computational cost compared to naively computing text embeddings for every attribute. In comparison to descriptors, PAC does use slightly more attributes for each category. However, this also implies that PAC's method of querying

| | CALTECH | DTD | FLOWERS | PETS |
|---|---|---|---|---|
| `{category_name} which has {description}` | 87.27 | 55.31 | 77.76 | 92.25 |
| `{category_name} which has {attribute} {description}` | 87.17 | 55.79 | **78.50** | 92.42 |
| `{category_name} with {description}` | 87.42 | 55.42 | 76.48 | 92.75 |
| `{category_name} with {attribute} {description}` | **87.48** | 56.27 | 75.60 | 92.69 |
| `{category_name} ({description})` | 87.32 | **57.65** | 76.97 | 91.25 |
| `{category_name} ({attribute} {description})` | 86.93 | 57.39 | 78.06 | 90.76 |
| `{category_name} with ({description})` | 86.76 | 56.01 | 75.97 | 92.12 |
| `{category_name} with ({attribute} {description})` | 86.73 | 56.32 | 76.24 | **92.85** |

Table 4: **Ablation Study: Different Text Prompt Formulations**. Results are produced with CLIP ViT-L/14 architecture. Although the optimal formulation may vary between datasets, we keep the formulation found on Oxford-Pets to ensure fairness of evaluation.

pair-specific attribute descriptions aggregates more discriminative information for each category. Experimental results substantiate the advantages of PAC in this regard.

**Evaluation on Coarse Label Datasets**. While the primary motivation behind PAC is to address model confusion among fine-grained classes, it is intriguing to explore how PAC performs on datasets with coarser labels[1], a challenge also addressed by CHiLS (Novack et al., 2023). Specifically, categories of these datasets can usually be further classified into other subclasses. For example, *Bottle* category in Office-Home can be further classified into water bottle, wine bottle, etc. CHiLS (Novack et al., 2023)

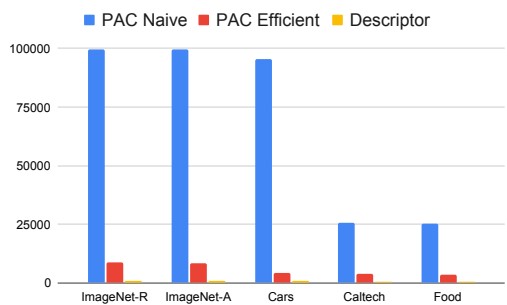

leverages existing hierarchies or LLMs to convert coarse category names into a list of subclasses and perform standard zero-shot inference across all subclasses. The final predicted subclass is inversely mapped to original coarse class as the final prediction.

As shown in Table 5, it is interesting to see that PAC can even outperform CHiLS on some of these datasets. We speculate that PAC's method of querying LLMs, which involves asking for pair-specific attribute descriptions, implicitly broadens the context of categories. In the case of coarse labels, these attributes may prove to be even more effective than subclass names, as this still relies on the model's understanding of subclass names (examples are provided in Appendix A.3).

| | OFFICEHOME | FOOD-101 | EUROSAT | RESISC45 |
|---|---|---|---|---|
| CLIP Radford et al. (2021) | 88.8 | 93.9 | 62.1 | 72.6 |
| CHiLS (Novack et al., 2023) | 88.8 | 93.8 | 62.4 | **72.7** |
| PAC (ours) | **89.6** | **94.2** | **65.2** | 71.5 |

Table 5: **Results on Datasets with Coarse Labels**. Results are produced with CLIP ViT-L/14@336 architecture. PAC can even ourperform CHiLS on some datasets with coarse labels.

## 5 CONCLUSION AND FUTURE WORK

We introduced Pairwise Attribute Contrasting (PAC), a framework designed to enhance the zero-shot classification performance of vision-language models. PAC leverages Large Language Models (LLMs) to generate pair-specific attribute descriptions and reformulates the inference as a pairwise binary classification voting problem. Experimental results validate PAC's effectiveness across various datasets. PAC does entail additional computational complexity. As such, we propose an efficient implementation that is significantly faster than the naive approach. Future research could explore alternative strategies for attribute selection from LLMs to further reduce computational costs.

---

[1]Note that Food-101, EUROSAT, and RESISC45 datasets have typically been characterized as fine-grained datasets in prior literature (Radford et al., 2021; Allingham et al., 2023). However, they are described as having coarse-label classes by CHiLS (Novack et al., 2023). In our opinion, these classes lean more towards the coarse-label side, as their categories often have the potential for further classification into subclasses.

**Reproducibility Statement.** To ensure reproducibility, we provide sufficient details on how we query LLMs for pair-specific attribute descriptions in Section 3.2 and Appendix A.1. We also thoroughly discuss how we formulate text prompts and examples of few alternatives in Table 4. Since we evaluate on open-sourced vision-language models, these models should be generally available to the public. Therefore, we believe our work can be successfully reproduced by the community. We will also release our code upon acceptance.

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

# A  APPENDIX

## A.1  DETAILS OF PROMPTING LLMS

We use the OpenAI backend *gpt-3.5-turbo* as our LLM for all experiments. As discussed in Section 3.2, we use the following prompt to query GPT-3.5:

```
Q: What visual attributes can differentiate between {category A}
   and {category B}?
A:
  - Attribute 1:
     - Category i: Description for Attribute 1
     - Category j: Description for Attribute 1
  - Attribute 2:
     - Category i: Description for Attribute 2
     - Category j: Description for Attribute 2
  ...
```

For each pair, we ask for 5 distinct attributes with descriptions. To better format the outputs from LLMs, we provide one example in the desired format:

```
For example, given Abyssinian and Sphynx:
- Fur Length:
   - Abyssinian: Short, dense coat
   - Sphynx: Hairless or very little hair
- Eye Shape:
   - Abyssinian: Almond-shaped eyes
   - Sphynx: Large round eyes
```

The knowledge in this tiny example is also returned by GPT-3.5 with the above query without providing formatting examples. In other words, we do not manually inject any knowledge even through this tiny example. We also experiment with other choices of LLMs such as *gpt-3.5-turbo-16k* and *gpt-4* yet do not observe significantly better results with these more advanced models, as shown in Table A. We also experimented with legacy GPT-3 models such as *text-davinci-002* and *text-davinci-003*, however, these models perform worse in formatting pair-specific attributes with the desired format. We speculate this is because of the auto-regressive training approach. Considering the cost of these models are even higher ($0.02 / 1K tokens versus $0.0015 / 1K tokens for *gpt-3.5-turbo*) and will be deprecated in early 2024, we use *gpt-3.5-turbo* as the default choice for our approach.

|  | CIFAR-10 | DTD | PETS | FLOWERS | FOOD |
|---|---|---|---|---|---|
| *gpt-3.5-turbo* | 95.49 | 56.64 | 94.43 | 78.48 | 93.59 |
| *gpt-3.5-turbo-16k* | 95.34 | 57.44 | 94.73 | 79.34 | 93.51 |
| *gpt-4* | 95.32 | 57.87 | 94.41 | 79.78 | 93.56 |

Table A: **Results of PAC with pair-specific attributes using different LLMs**.

## A.2  ZERO-SHOT ACCURACY ON INDIVIDUAL DATASETS

We report average evaluation results over 5 ImageNet variations and 11 fine-grained classes in Table 3. In this section, we provide zero-shot accuracy of PAC using different visual encoder architectures on individual datasets.

|  | CALTECH | CARS | C10 | C100 | DTD | EURO | FOOD | FLOWERS | PETS | RESISC | SUN |
|---|---|---|---|---|---|---|---|---|---|---|---|
| RN50 | 80.16 | 54.77 | 73.93 | 42.02 | 43.67 | 31.44 | 80.07 | 67.53 | 85.93 | 54.18 | 60.73 |
| RN101 | 84.66 | 61.90 | 81.58 | 50.36 | 45.95 | 30.63 | 83.80 | 67.91 | 86.10 | 55.27 | 60.86 |
| ViT-B/16 | 86.29 | 64.83 | 89.81 | 67.51 | 49.46 | 57.91 | 89.22 | 72.56 | 89.88 | 65.06 | 67.17 |
| ViT-L/14 | 89.15 | 76.52 | 95.49 | 77.17 | 57.39 | 60.84 | 93.59 | 78.76 | 94.43 | 71.00 | 70.12 |

Table B: **Results of PAC on individual fine-grained datasets with different CLIP backbones**.

| | IMAGENET-VAL | IMAGENET-V2 | IMAGENET-SKETCH | IMAGENET-ADVERSARIAL | IMAGENET-RENDITION |
|---|---|---|---|---|---|
| RN50 | 59.83 | 58.93 | 35.57 | 24.76 | 60.63 |
| RN101 | 62.15 | 62.06 | 41.26 | 29.77 | 68.54 |
| ViT-B/16 | 68.1 | 67.68 | 48.44 | 50.69 | 77.62 |
| ViT-L/14 | 75.03 | 74.4 | 59.17 | 71.03 | 87.68 |

Table C: **Results of PAC on ImageNet variations with different CLIP backbones**.

### A.3 PAIR-SPECIFIC ATTRIBUTE DESCRIPTIONS FOR COARSE LABELS

We present some examples from EuroSat (Helber et al., 2019) and RESISC (Cheng et al., 2017), both of which have been recognized by CHiLS (Novack et al., 2023) as datasets characterized by coarse labels. In these datasets, class names often encompass multiple subclasses, leading to challenges in constructing informative text prompts.

As depicted in Figure A, while most category descriptions are meaningful, some attribute names, like "presence of hoops" and "presence of seating" may not be suitable for inclusion in text prompts because they tend to disrupt the natural flow of a sentence when integrated into the prompt. Additionally, attributes like "texture" are typically not employed to describe categories related to beach and harbor scenes. Consequently, when devising text prompts for categories within these datasets, we intentionally omit attribute names, as detailed in Section 4.2.

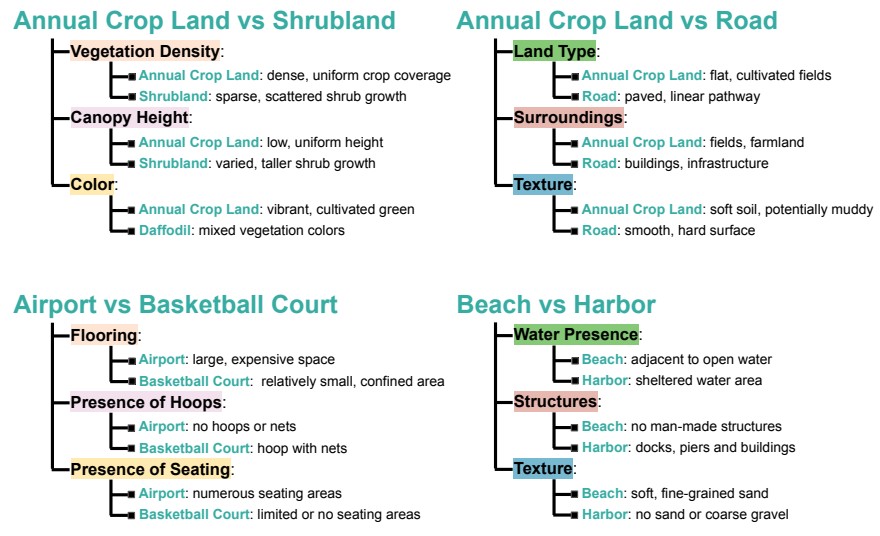

Figure A: **Example of Pair-specific Attribute Descriptions on Datasets with coarse labels**.

