# OpenReview forum: "Zero-shot Visual Recognition via Pairwise Attribute Contrasting"
_ICLR.cc/2024/Conference — ICLR 2024 Conference Withdrawn Submission_

### Official Review · Reviewer_udyw · 2023-10-27

**Soundness:** 3 good
**Presentation:** 3 good
**Contribution:** 2 fair
**Rating:** 5
**Confidence:** 4

**Summary:**

The paper addresses the zero-shot image classification problem, building upon the work of Menon & Vondrick (2022) that utilized isolated category inquiries in prompting language models. In this extension, the authors propose a novel approach that leverages language models to generate pair-specific attribute descriptions for category pairs and subsequently employs a voting procedure akin to one-versus-one SVM. The evaluation encompasses extensive experiments on 18 benchmark datasets.

**Strengths:**

One of the strengths of this work is the rigorous experimentation. The authors have thoroughly evaluated their proposed method across multiple datasets, utilizing various backbone models. While the improvements over the state-of-the-art results are reported, it's worth noting that the significance of these improvements is relatively modest.

**Weaknesses:**

One of the concerns with this paper relates to its novelty. Although the paper extends the approach proposed by Menon & Vondrick (2022), the core novelty primarily stems from a change in the prompt to acquire more distinctive category descriptions. While this change is interesting, it also brings a substantial increase in computational complexity, transitioning from a linear to a quadratic scale. Despite the proposed scheme to mitigate this complexity, the fundamental nature of the method remains computationally more demanding, which may limit its scalability.

**Questions:**

Please address the novelty and the scalability concerns.

---

### Official Review · Reviewer_3Fkf · 2023-10-28

**Soundness:** 2 fair
**Presentation:** 2 fair
**Contribution:** 2 fair
**Rating:** 5
**Confidence:** 3

**Summary:**

This paper proposes Pairwise Attribute Contrasting (PAC), a zero-shot inference framework for vision-language models. PAC prompts LLMs to provide visual attributes to distinguish category pairs and aggregate the pairwise comparisons into a single classification using a voting procedure. The paper conducts experiments on 18 benchmark datasets to verify the effectiveness of the method.

**Strengths:**

- The author's research motivation is easily understandable.
- The Pairwise Attribute Contrasting (PAC) for enhancing discriminative information is reasonable.

**Weaknesses:**

- Although the author has optimized the method, the computational cost of the optimized method (PCA Efficient) is still nearly dozens of times greater than the Descriptor, but the performance improvement relative to the Descriptor is not significant.(The computational cost shown in the picture is not clear, it is recommended to mark the specific value)
- The supplementary material states that each pair provides 5 distinct attributes with descriptions. How is the number of attributes determined? Is it dynamic or fixed?
- More ablation experiments are needed. The paper directly uses pairwise description contrasting for some coarse label datasets. For other datasets, how effective is the simple pairwise description contrasting? Is it necessary to use attributes?
- The paper contains some minor errors:
    - Inconsistent descriptions. For instance, "remove {attribute_name}"(the second to last paragraph on page 8) does not correspond to the description in Table 4.
    - The layout of figures. For instance, the figure on page 6 has inconsistent spacing for category names (some use spaces, others use line breaks). And the figure on page 9 is not placed near the corresponding text description and the axis units are missing.

**Questions:**

Please see the weaknesses.

---

### Official Review · Reviewer_Zv8d · 2023-10-30

**Soundness:** 2 fair
**Presentation:** 3 good
**Contribution:** 1 poor
**Rating:** 3
**Confidence:** 4

**Summary:**

This paper introduces a novel approach that leverages Large Language Models (LLM) to obtain prior attribute information for specific classes. It then utilizes this information to measure the similarity between class attributes and images, enabling recognition based on similarities within the CLIP space. To extract valid attribute information, the paper introduces Pairwise Attribute Contrasting (PAC), which instructs the LLM to identify key attributes between given classes.

**Strengths:**

1. Clarity and Organization: This paper is exceptionally well-written and thoughtfully organized, making it easy for readers to follow the author's reasoning and methodology.

2. Clear Motivation: The paper's motivation is not only clear but also compelling, underscoring the significance of the problem addressed.

3. Effective Simplicity: The proposed method, while seemingly simple, proves to be highly effective in achieving its objectives, demonstrating elegant problem-solving.

**Weaknesses:**

1. The novelty in this paper appears to be somewhat incremental, as it primarily builds upon the methodin [1]. The key differentce between this paper and [1] lies in the way of attribute extraction.
While [1] relies on Large Language Models (LLM) to provide concise attributes or semantic descriptions, this paper advances the field by tasking LLM with generating more detailed attributes through class comparisons.
 It's worth noting that the shift in strategy for extracting prior attribute information from LLM, while noteworthy, may not represent a significant breakthrough.
However, it's important to acknowledge that this paper comes with an increased computational cost, and the resulting performance gains are somewhat limited when compared to [1].

2. Given the limited extent of the performance improvements, it is advisable for the author to delve further into the analysis. Similar to [1], the author should consider including instances of failure cases generated by LLM. This addition would offer valuable insights for a more thorough understanding of the method's strengths and limitations.

[1] Sachit Menon and Carl Vondrick. Visual classification via description from large language models. arXiv preprint arXiv:2210.07183, 2022

**Questions:**

See `Weaknesses' above.

---

### Official Review · Reviewer_qUu9 · 2023-11-02

**Soundness:** 2 fair
**Presentation:** 2 fair
**Contribution:** 2 fair
**Rating:** 5
**Confidence:** 3

**Summary:**

This paper proposes Pairwise Attribute Contrasting (PAC) to enhance the zeroshot classification performance of vision-language models. PAC leverages LLMs to generate pair-specific attribute descriptions and reformulates the inference as a pairwise binary classification voting problem. The effectiveness is verified on 18 datasets.

**Strengths:**

1. The motivation for the paper is sound and simple.
2. PAC shows consistent improvement on 18 benchmark datasets over other strong baselines across various model architectures.
3. Zero-shot inference is popular research recently, and the authors' research contributes to the development of this community.

**Weaknesses:**

1. The performance improvement of the method is weak compared to state-of-the-art methods, and the number of papers compared is limited.
2. For ICLR, the contribution of this work may not be enough.
3. Although methods have been proposed to reduce the computational complexity, they are still unacceptable.
4. There are some minor detail issues (for example, the figures lack descriptions on pages 6 and 9).
5. Recent research based on learnable prompts is interesting. Although the authors introduced them in related work, no quantitative performance comparison was performed.
6. How to define attribute? What is the difference between attribute? More detailed instructions should be provided.
7. Some details, such as Figure and Fig should be unified. Please check carefully with the author.

**Questions:**

See the Weaknesses.